# Evaluation of an intervention to provide brief support and personalized feedback on food shopping to reduce saturated fat intake (PC-SHOP): A randomized controlled trial

**Carmen Piernas** \*, **Paul Aveyard**, **Charlotte Lee**, **Melina Tsiountsioura**,
**Michaela Noreik**, **Nerys M. Astbury**, **Jason Oke**, **Claire Madigan**, **Susan A. Jebb**

Nuffield Department of Primary Care Health Sciences, University of Oxford, Oxford, United Kingdom

\* carmen.piernas-sanchez@phc.ox.ac.uk

## Abstract

pmed.1003385

Population Center, UNITED STATES

**Data Availability Statement:** This study includes
potentially sensitive participant information data

### Background

Guidelines recommend reducing saturated fat (SFA) intake to decrease cardiovascular dis-
ease (CVD) risk, but there is limited evidence on scalable and effective approaches to
change dietary intake, given the large proportion of the population exceeding SFA recom-
mendations. We aimed to develop a system to provide monthly personalized feedback and
healthier swaps based on nutritional analysis of loyalty card data from the largest United
Kingdom grocery store together with brief advice and support from a healthcare professional
(HCP) in the primary care practice. Following a hybrid effectiveness-feasibility design, we
tested the effects of the intervention on SFA intake and low-density lipoprotein (LDL) choles-
terol as well as the feasibility and acceptability of providing nutritional advice using loyalty
card data.

### Methods and findings

The Primary Care Shopping Intervention for Cardiovascular Disease Prevention (PC-
SHOP) study is a parallel randomized controlled trial with a 3 month follow-up conducted
between 21 March 2018 to 16 January2019. Adults ≥18 years with LDL cholesterol >3
mmol/L (*n* = 113) were recruited from general practitioner (GP) practices in Oxfordshire and
randomly allocated to "Brief Support" (BS, *n* = 48), "Brief Support + Shopping Feedback"
(SF, *n* = 48) or "Control" (*n* = 17). BS consisted of a 10-minute consultation with an HCP to
motivate participants to reduce their SFA intake. Shopping feedback comprised a personal-
ized report on the SFA content of grocery purchases and suggestions for lower SFA swaps.
The primary outcome was the between-group difference in change in SFA intake (% total
energy intake) at 3 months adjusted for baseline SFA and GP practice using intention-to-
treat analysis. Secondary outcomes included %SFA in purchases, LDL cholesterol, and fea-
sibility outcomes. The trial was powered to detect an absolute reduction in SFA of 3%
(SD3). Neither participants nor the study team were blinded to group allocation.

collected by the research team from March 2018 to January 2019, which includes information on the clinical measurements, health conditions, medications, and dietary intake measurements (including the primary outcome measure). Anonymized version of this data is available from the corresponding author on reasonable request. This study also includes grocery shopping data from the collaborating supermarket, which were received weekly from March 2018 to January 2019. Because of contractual restrictions to the use of the grocery shopping data, it is not possible to make these data openly available outside this project but any queries can be directed to the University Research Services (carly. banner@admin.ox.ac.uk) and the Clinical Trials & Research Governance and the Health Research Authority that provided ethical approval (ctrg@admin.ox.ac.uk; nrescommittee. southcentral-oxfordc@nhs.net) Grocery shopping data were supplemented with a dataset containing the nutritional composition of foods provided by a commercial organization (Brand View). Further use of this data must be negotiated with the data owners Brand View (support@brandview.com).

**Funding:** The study was funded by the National Institute of Health Research (NIHR) Collaboration for Leadership in Applied Health Research and Care, Oxford as well as the School of Primary Care Research. PA and SAJ are NIHR Senior Investigators and funded by the Oxford Biomedical Research Centre and CLAHRC. NA is funded by Oxford Biomedical Research Centre. Grocery shopping data were provided by Tesco. Nutrient composition data for the products sold in grocery stores were provided by Brand View. The funders had no role in study design, data collection and analysis, decision to publish, or preparation of the manuscript.

**Competing interests:** The authors have declared that no competing interests exist.

**Abbreviations:** BMI, body mass index; BS, brief support; CVD, cardiovascular disease; EI, energy intake; GP, general practitioner; HCP, healthcare professional; LDL-C, low-density lipoprotein cholesterol; MUFA, monounsaturated fatty acids; NHS, National Health System; PUFA, polyunsaturated fatty acids; RCT, randomized controlled trial; SF, shopping feedback; SFA, saturated fatty acids; TE, total energy.

A total of 106 (94%) participants completed the study: 68% women, 95% white ethnicity, average age 62.4 years (SD 10.8), body mass index (BMI) 27.1 kg/m$^2$ (SD 4.7). There were small decreases in SFA intake at 3 months: control = −0.1% (95% CI −1.8 to 1.7), BS = −0.7% (95% CI −1.8 to 0.3), SF = −0.9% (95% CI −2.0 to 0.2); but no evidence of a significant effect of either intervention compared with control (difference adjusted for GP practice and baseline: BS versus control = −0.33% [95% CI −2.11 to 1.44], $p = 0.709$; SF versus control = −0.11% [95% CI −1.92 to 1.69], $p = 0.901$). There were similar trends in %SFA based on supermarket purchases: control = −0.5% (95% CI −2.3 to 1.2), BS = −1.3% (95% CI −2.3 to −0.3), SF = −1.5% (95% CI −2.5 to −0.5) from baseline to follow-up, but these were not significantly different: BS versus control $p = 0.379$; SF versus control $p = 0.411$. There were small reductions in LDL from baseline to follow-up (control = −0.14 mmol/L [95% CI −0.48, 0.19], BS: −0.39 mmol/L [95% CI −0.59, −0.19], SF: −0.14 mmol/L [95% CI −0.34, 0.07]), but these were not significantly different: BS versus control $p = 0.338$; SF versus control $p = 0.790$. Limitations of this study include the small sample of participants recruited, which limits the power to detect smaller differences, and the low response rate (3%), which may limit the generalisability of these findings.

## Conclusions

In this study, we have shown it is feasible to deliver brief advice in primary care to encourage reductions in SFA intake and to provide personalized advice to encourage healthier choices using supermarket loyalty card data. There was no evidence of large reductions in SFA, but we are unable to exclude more modest benefits. The feasibility, acceptability, and scalability of these interventions suggest they have potential to encourage small changes in diet, which could be beneficial at the population level.

## Trial registration

ISRCTN14279335.

Author summary

### Why was this study done?

- Cardiovascular disease (CVD) is the leading cause of death in the UK and is strongly influenced by diet composition. Reducing the intake of saturated fat (e.g., fats from animal sources such as butter or meat), mostly by swapping some key foods in the diet for others that are lower in saturated fat (SFA), can help reduce the "bad" low-density lipoprotein cholesterol (LDL-C) in the blood, and reduce the risk of CVD.

- Previous studies have achieved success either by providing particular foods to people or by giving them intensive support and advice from nutrition specialists. Currently, there are no practical interventions shown to help large numbers of people improve their diet to reduce the amount of saturated fat they eat.

## What did the researchers do and find?

- In this study, we developed a system to provide regular information on the saturated fat content of food purchases and suggest healthier swaps using loyalty card data from the UK largest grocery store. Participants received brief oral and written advice from a healthcare professional (HCP) at their general practitioner (GP) practice alone or in combination with personalized feedback on their food shopping. Our primary aim was to test whether this approach was effective to decrease saturated fat intake compared with usual care, which does not involve any specific advice. We also compared changes in LDL-C and the quality of the grocery shopping. We recruited adults who had a blood test showing they had raised LDL-C from GP practices in Oxfordshire (UK), and we followed them for an average of 3.8 months.

- We found small decreases in SFA intake as well as the SFA content of food purchases and reductions in LDL-C, but these changes were not significantly different from those observed in the control group. Participants reported positive feedback regarding the brief advice and the personalized feedback on their food shopping, which they received monthly throughout the study.

## What do these findings mean?

- Previous studies have shown that self-monitoring and feedback are effective strategies to help people change their behavior. In this study, we were able to use data from supermarket loyalty cards to provide regular feedback and healthier swaps to help people improve the quality of their grocery shopping. Participants valued and used this information together with the brief advice received from primary care practitioners to reduce their saturated fat intake.

- The trial was designed to detect a clinically significant difference in SFA intake between groups of 3%, and the intervention did not achieve such large effects. However, modeling studies suggest that just replacing 1% of saturated fat with polyunsaturated fat can potentially lead to an 8% reduction in CVD events. With future development and testing, this may be an intervention that could be offered by supermarkets to achieve small improvements in diet with population-level health benefits.

## Introduction

A diet high in saturated fatty acids (SFA) elevates low-density lipoprotein cholesterol (LDL-C) and increases the risk of cardiovascular disease (CVD) [1–3]. SFA intake in the UK population (13.5% total energy) and in other higher income countries remains higher than the dietary recommendation of <10% of the total energy intake, and it is estimated that >70% of the UK population exceeds this level [4, 5]. Systematic reviews and recent trials have shown that reductions in dietary SFA intake can lower LDL-C, but this has only been achieved with specialist staff and high-intensity behavioural support as well as with the provision of appropriate food substitutes in place of higher SFA products [6–9]. There is evidence that brief behavioral

interventions delivered in routine primary care can help people engage in weight loss programs with the potential to achieve population-level impact [10]. However, there is limited evidence of brief interventions that are effective for improving the nutritional quality of diet and sufficiently scalable for routine delivery in primary care settings by generalist staff or at the population level.

Food purchased to prepare and eat at home comprises the majority of food consumed in most countries, and interventions targeting the nutritional quality of food purchases could improve diet quality, especially among those motivated to change [11]. Many foods high in SFA, such as meat, dairy, ready meals, cakes, and biscuits, are bought to consume at home [12]. However, previous evidence suggests that information alone on the nutrient content of foods rarely leads to sustained dietary change, and other support may be needed to change dietary behaviours [13]. Interventions providing tailored dietary advice, personalized feedback, and encouraging self-monitoring appear promising [13–17]. Interventions delivered in grocery stores can also support individual behavior change. For example, in-store education increases fruit and vegetable purchases [18], while recommending healthier swaps at the point of purchase, including lower SFA alternatives, has been shown to improve the nutritional quality of food selection or purchasing [19–21].

Building on what is known, but with a specific focus on scalability and sustainability, we developed a behavioral intervention to reduce SFA consumption using data collected through a supermarket loyalty card. Loyalty cards recording food purchases are operated by many major retailers and offer the opportunity to provide personalized dietary interventions at scale. We tested the intervention among patients in primary care with raised LDL-C who received additional motivation to change and brief advice from a healthcare professional (HCP) in a short counselling session. Following a hybrid effectiveness-feasibility design [22], the aim of this study was to seek preliminary evidence of the effectiveness and feasibility of these interventions to reduce SFA intake and lower CVD risk.

## Methods

### Trial design

The Primary Care Shopping Intervention for Cardiovascular Disease Prevention Study (PC-SHOP) was a randomized, 3-arm parallel controlled open-label trial with blinded assessment of the primary outcome. Participants were individually randomized to either the control group or one of 2 active interventions for 3 months after giving written informed consent to participate in the study.

This study was reviewed and approved by the National Health Service Health Research Authority (HRA) Research Ethics Committee (Ref: 17/SC/0168). The trial protocol includes further information regarding the trial design and interventions [23]. A protocol and statistical analysis plan were prospectively registered at ISRCTN14279335 in September 2017 and March 2019, respectively, with prospective changes introduced to reflect delays in the planned recruitment and follow-up dates as well as the addition of a qualitative substudy and a few exploratory outcomes (S1 Appendix).

### Trial setting and recruitment

Four primary care practices in Oxfordshire, UK, near the participating supermarket were selected so it was more likely that potential participants shopped at this store. General practices sent invite letters to patients who were aged ≥18 years and with a recorded LDL cholesterol >3.5 mmol/L or total cholesterol >5.5 mmol/L in the previous 5 years. Eligible participants were adults aged ≥18 years; with a baseline LDL ≥3 mmol/L on retesting; willing to make

changes to their diet in order to reduce CVD risk; who had responsibility for the majority of the household food/grocery shopping (e.g., complete at least half of their household shopping); who usually shop at the collaborating grocery store (at least every 2 weeks in store and/or online); with a loyalty card registered exclusively under their name before recruitment; with access and ability to use a computer with internet connection; and willing and able to give informed consent for participation in the study. During screening, we excluded people with a self-reported recent incident of CVD, recent or planned changes to lipid medications, or those taking part in other relevant research. Interested and potentially eligible people were invited to a baseline appointment with the study team at their general practice to discuss and agree to participate and check eligibility again. People who who failed to complete the 2 baseline dietary assessments were excluded (Fig 1, CONSORT diagram; S2 Appendix, List of inclusion and exclusion criteria).

## Randomisation and blinding

Participants were individually randomized in a 1:3:3 ratio, to either the control group or one of 2 active interventions. A computer-generated randomization sequence was generated by an independent statistician using fixed block sizes of 7 with random order. After enrollment, participants' allocation to each trial arm was revealed to the researchers using an online program (RedCap, https://www.project-redcap.org/) to ensure full concealment. The researchers needed to know allocation in order to generate personalized shopping feedback reports for participants randomized to the SF group.

The brief advice component of the interventions meant that it was impossible to fully blind HCPs, but they remained blind to which of the 2 active intervention groups the participant was allocated. Likewise, participants could not be blind to allocation, and there were insufficient research staff to blind them to allocation at follow-up when some of the clinical measures (secondary outcomes) were taken. The primary outcome was saturated fat intake collected through a web-based questionnaire, which individuals completed without involving the study team and was analyzed blind by an independent statistician.

## Interventions

Details of the active interventions and their theoretical framework can be found in the protocol [23]. In brief, HCPs received 1 hour of training plus materials from the study team to deliver the brief intervention. Participants allocated to the "Brief Support" (BS) or "Brief Support plus Shopping Feedback" (SF) intervention attended a single 10-minute appointment with a HCP (e.g., nurse or healthcare assistant) at their usual GP practice, which was intended to boost motivation and capability to enact dietary change. They informed participants about the effect of SFA on blood cholesterol and CVD risk and provided tips to reduce SFA, guided by the British Heart Foundation "Cut the Saturated Fat" chart [24, 25] and the National Health System (NHS) Choices website (provided in printed form).

Participants allocated to the SF intervention also received a report on the saturated fat in their household grocery store food purchases during the intervention period. The study team received data from the participating supermarket each week and generated and sent participants a personalized shopping report on purchases in the preceding period using supermarket loyalty card data, at baseline and at the end of the first, second, and third month of the intervention period. The shopping reports provided information on the mean weekly saturated fat content of their food shopping, identified 5 major contributors to SFA intake over the previous period, as well as suggestions for one-for-one swaps to foods containing less SFA but with similar functional characteristics. Participants were encouraged to use the monthly reports to

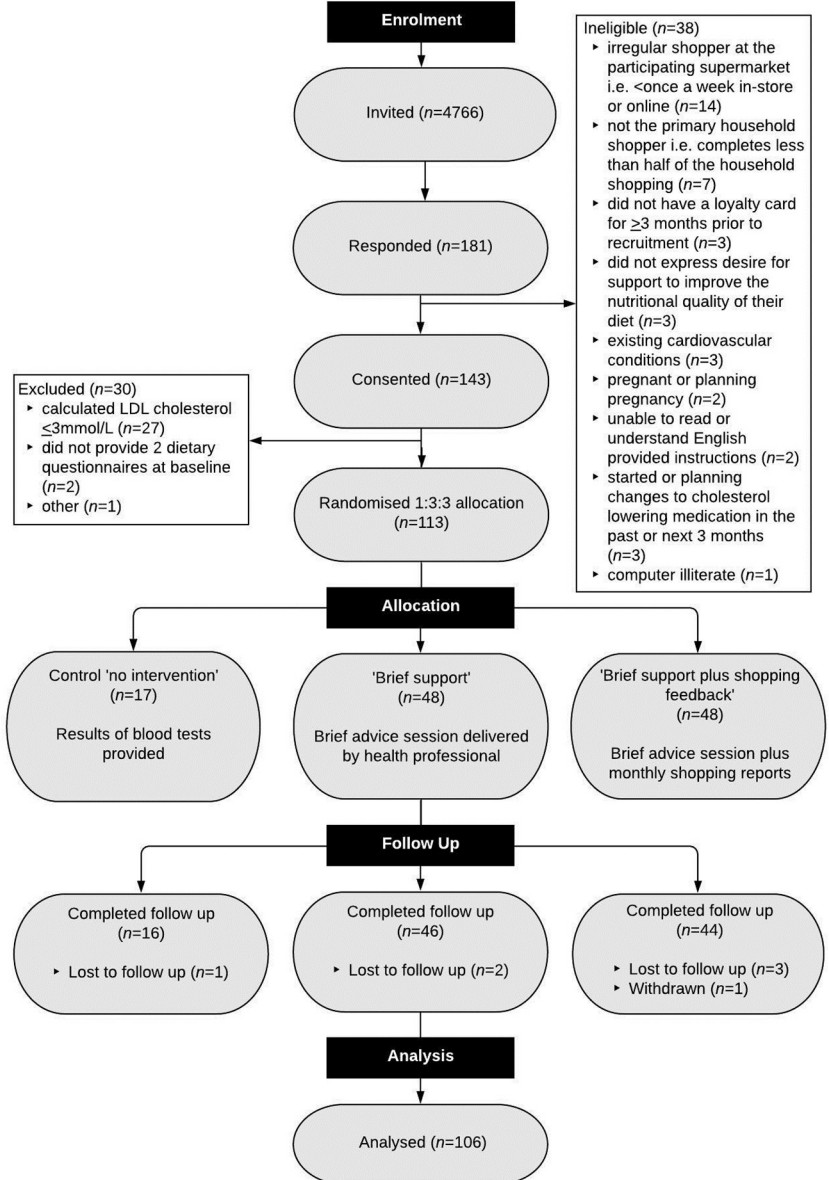

**Fig 1. CONSORT flowchart.** CONSORT, Consolidated Standards of Reporting Trials; LDL-C, low-density lipoprotein cholesterol.

guide their shopping list and to monitor their progress in reducing SFA. Details of the approach and algorithm to generate personalized shopping advice can be found elsewhere [23].

Participants in the control group received no intervention. They were informed of the results of their blood tests by post and invited to a further check after 3 months (end of follow-up period).

## Procedures

Participants were asked to complete a computerized 24-hour dietary recall prior to the baseline visit and a second one at the baseline visit. Participants provided their supermarket loyalty

> ## Box 1. Intervention components.
>
> **Brief advice session with a nurse or healthcare assistant at the GP practice**.
>
> 1. Single 10-minute appointment.
>
> 2. Written materials: British Heart Foundation "Cut the Saturated Fat" leaflet and NHS Choices information together with blood results.
>
> **Personalized shopping report**.
>
> 1. One report per month from baseline to follow-up (up to 4 reports in total, print form and email) containing information on weekly amount of saturated fat in shopping, top 5 food contributors to saturated fat plus lower saturated fat swaps to foods with similar functional characteristics.

card number and consent to access information on all food purchases from 6 months before baseline until the end of the study. Trained researchers measured weight, height, blood pressure, and collected demographic information and relevant medical history and medications. A fasting capillary blood sample was collected and analyzed using a point of care device (Alere Cholestech LDX) to determine total cholesterol, high-density lipoprotein cholesterol (HDL-C), LDL-C, and triglycerides. After randomization, participants in BS and SF groups met with a HCP for the brief advice session.

A single follow-up visit took place 3 months later, repeating the baseline measures (except height) and including an additional questionnaire assessing the acceptability of the intervention.

## Dietary intake and shopping data

Individual self-reported dietary intake was assessed using the validated questionnaire, the Oxford Biobank WebQ, which collects information on the quantities of all foods and beverages consumed over the previous day [26]. The quantity (g) of each food and beverage consumed was calculated by multiplying the average portion size [27] by the number of portions consumed. Nutrient intakes including total energy and SFA in grams were automatically calculated from a UK-specific food composition table [28] linked to the WebQ. We calculated the percentage of energy intake (%EI) from each macronutrient as well as the average daily intakes across the 2 days that were recorded at baseline and again at follow-up. We also calculated the energy contribution (%EI) of those food groups that are important sources of SFA: cakes, biscuits and desserts, meat (including red and processed meat), poultry, cheese, yogurt, spreads, and salty snacks.

Shopping data on all recorded food purchases from 6 months before baseline and during the study were also used to evaluate the effect of the intervention given the high correlations found with dietary intake measurements [29]. Shopping data were linked to a database of approximately 20,000 food and drink products, which included information on nutrient composition, price, and volume per unit (Brand View Limited). We calculated the total basket (kcal and g) and SFA, total fat, sugars, salt, and fiber as well as the energy contribution from food groups aligned to the WebQ food groups, plus ready-to-eat meals. We calculated averages before and after baseline to obtain the percentage of energy from SFA and percentage from

total basket energy (%TE) from grocery store purchases, as well as percentage of other macro-nutrients, and the percentage contribution of food groups before and after the intervention.

## Outcomes

The primary outcome was the change in reported saturated fat intake (%SFA) measured on 2 days at baseline and again at follow-up. The full list of prespecified secondary and nonefficacy outcomes included changes from baseline to follow-up in SFA (kcal), total energy intake (kcal), total fat (kcal, %EI), total sugars (kcal, %EI), fiber (g/100 kcal), and key food groups (%EI). Other exploratory outcomes from the dietary measurements were changes in total mono-unsaturated fatty acids (MUFA) and polyunsaturated fatty acids (PUFA) intakes between base-line and follow-up. Prespecified secondary outcomes from food purchases included changes from baseline to follow-up in SFA (%TE; kcal per £), total fat and sugars (%TE; kcal per £), salt (g/100 g), cost (£/week and £/Kg), energy density (kcal/g), and key food groups (%TE) expressed as percentages to be comparable between the diet and the shopping data. We also assessed changes in LDL-C, HDL-C, total cholesterol, and triglycerides.

For the process evaluation, we assessed the feasibility of recruitment and follow-up by mea-suring the number of participants who accepted the invitation and consented to take part as well as the number of participants that completed follow-up. We measured acceptability of the intervention through nonvalidated questionnaires completed at baseline and follow-up (rated on a scale 1–5 from least to most helpful, S3 Appendix). The fidelity of the intervention deliv-ery by the HCP was measured through the analysis of audio-taped sessions.

## Statistical analysis

We powered the study to detect a difference in %SFA reduction of 3% (3% SD) between each intervention group and control, which was considered plausible based on previous interven-tions [8, 9] and likely to be beneficial at the individual level to reduce LDL cholesterol [3]. We calculated a total sample of 112 (16 control, 48 BS, and 48 SF group) using intention-to-treat analysis with 90% power, 10% attrition, and statistical adjustment for multiple comparisons (using 2-sided $\alpha = 0.01$ in the primary outcome analysis). Using a 1:3:3 allocation ratio, we cal-culated a sample of 48 participants in each active intervention arms to have 90% power to detect a further difference in %SFA intake between BS and SF interventions of 2% (3% stan-dard deviation). All the other tests were performed at a 5% two-sided significance level, Stata 14 was used for the analyses.

After the statistical analysis plan was approved by the trial management group and an inde-pendent statistician, the database was locked, and the statistical analysis plan was implemented without changes to the primary or secondary outcome analyses. The results of the trial were computed as comparative summary statistics by group (difference in means between baseline and follow-up, 95% CIs), and the primary comparisons between groups were analyzed with a linear regression model with adjustment for GP practice and baseline values. Prior to analysis, assumptions of linear regression (normality of residuals, homogeneity of variances, and outli-ers) were tested and met. Information on the acceptability of the intervention was collected by questionnaire and summarized by presenting the frequencies of each response.

We performed prespecified sensitivity analyses to assess the impact of missing data on the primary outcome, including using baseline observation carried forward, as well as excluding people with only one dietary recall or excluding specific days with reported dietary intakes reflecting implausible habitual intake (<500 kcal/day or >3,500 kcal/day [30]). We also excluded people who received their intervention session with the HCP >45 days from baseline, as well as those affected by changes in relevant medications (e.g., statins) during the study. We

performed a post hoc analysis adjusting for body mass index (BMI) to account for imbalance in the baseline measures. We also assessed whether the effect of the interventions differed by sociodemographic status in prespecified exploratory subgroup analyses.

## Results

### Baseline characteristics

Participants were recruited from 4 GP practices in Oxfordshire between 21 March 2018 and 10 October 2018. A final sample of 113 were randomized to either "No intervention" ($n = 17$), "BS" ($n = 48$) or "SF" ($n = 48$). Follow-up was completed on 16 January 2019, and 106 (94%) participants completed the study after an average of 115 days (SD 26, minimum 77, maximum 208) (Fig 1).

Participants were mostly women (68%), of white ethnicity (95%), with average age of 62.4 years (SD 10.8), and BMI 27.1 kg/m$^2$ (SD 4.7) (Table 1). Fourteen percent of participants reported a family history of CVD; 24% had hypertension, and 2% were taking lipid-lowering medication.

### Changes in reported dietary intake

Change in mean saturated fat intake (%EI) at 3 months (primary outcome) was −0.1% (95% CI −1.8 to 1.7) in the control group, −0.7% (95% CI −1.8 to 0.3) in BS, and −0.9% (95% CI −2.0 to 0.2) in SF (Fig 2, **Tables A-B in** S4 Appendix). There was no statistically significant difference in either intervention group compared with control (BS versus control = −0.33 [95% CI −2.11, 1.44] $p = 0.709$; SF versus control = −0.11% [95% CI −1.92, 1.69] $p = 0.901$; adjusted for GP practice and baseline value) or between intervention groups (S1 Table B).

There were no statistically significant differences between the intervention and control groups in total fat, total daily EI, sugars, dietary fiber (secondary outcomes), or PUFA and MUFA intakes (exploratory outcomes) (**Tables A-B in** S4 Appendix). Mean total fat intakes (kcal) and daily EIs (kcal) were reduced from baseline to follow-up among the intervention groups: BS = −109.8 kcal (95% CI −214 to −5.6); SF = −79.4 kcal (95% CI −186 to 27.2); and BS = −176.9 kcal (95% CI −393.1 to 39.2); SF = −106.2 kcal (95% CI −327.2 to 114.9), respectively, although these changes were not significant between intervention groups compared with control (Tables A-B in S1 Appendix).

Changes in the intakes of key food groups (%EI) were not statistically significantly different between intervention and control, but there were nonsignificant reductions from baseline to follow-up among the intervention groups (Fig 3, **Tables A-B in** S4 Appendix): cakes, biscuits, and desserts (BS = −2.6% [95% CI −5.7 to 0.4], SF = 0.5% [95% CI −2.7 to 3.6]); meat (BS = −2.3% [95% CI −4.5 to −0.2], SF = 0.0% [95% CI −2.2 to 2.2]); higher fat cheese (BS = −0.9% [95% CI −1.9 to 0.2]; SF = −1.3% [95% CI −2.3 to −0.2]), higher fat yogurt (BS = 0.2% [95% CI −0.4 to 0.8], SF = −0.8% [95% CI −1.4 to −0.2]), and higher fat spread (BS = −0.8% [95% CI −1.8 to 0.1]; SF = −1.3% [95% CI −2.2 to −0.3]).

### Changes in food purchasing

Changes in SFA (%TE) from food purchases at 3 months (secondary outcome) were −0.5% (95% CI −2.3 to 1.2) in the control group, −1.3% (95% CI −2.3 to −0.3) in BS, and −1.5% (95% CI −2.5 to −0.5) in SF (Fig 2, **Tables C-D in** S4 Appendix), but there were no statistically significant differences between the groups. The adjusted difference in %SFA in the total basket was −0.8% (95% CI −2.5 to 0.9) $p = 0.379$ between BS versus control; and −0.7% (95% CI −2.4 to 1.0) $p = 0.411$ between SF versus control. There were no statistically significant differences

**Table 1. Baseline characteristics of participants assigned to interventions or control.**

| | Total (n = 113) | | Control (n = 17) | | Brief Support (n = 48) | | Brief Support plus Shopping Feedback (n = 48) | |
|---|---|---|---|---|---|---|---|---|
| | Mean/n | SD/% | Mean/n | SD/% | Mean/n | SD/% | Mean/n | SD/% |
| Age, years, mean (SD) | 62.4 | 10.8 | 62.9 | 11.2 | 64.7 | 9.2 | 59.9 | 11.7 |
| Gender, female | 77 | 68.1 | 11 | 64.7 | 34 | 70.8 | 32 | 66.7 |
| BMI, kg/m², mean (SD) | 27.1 | 4.7 | 26.0 | 5.1 | 26.8 | 4.0 | 27.8 | 5.2 |
| **BMI categories** | | | | | | | | |
| Normal weight (18.5–24.9) | 38 | 33.6 | 7 | 41.2 | 14 | 29.2 | 17 | 35.4 |
| Overweight (25–29.9) | 53 | 46.9 | 8 | 47.1 | 27 | 56.3 | 18 | 37.5 |
| Obesity (≥30) | 22 | 19.5 | 2 | 11.8 | 7 | 14.6 | 13 | 27.1 |
| **Blood pressure, mean (SD)** | | | | | | | | |
| Systolic, mmHg | 131.5 | 17.5 | 130.5 | 11.6 | 130.0 | 19.6 | 133.3 | 17.2 |
| Diastolic, mmHg | 79.2 | 9.8 | 77.4 | 8.1 | 77.2 | 10.8 | 81.8 | 8.9 |
| **Smoking** | | | | | | | | |
| Current | 6 | 5.3 | 1 | 5.9 | 2 | 4.2 | 3 | 6.3 |
| Ex-smoker | 37 | 32.7 | 5 | 29.4 | 17 | 35.4 | 15 | 31.3 |
| Never | 67 | 59.3 | 10 | 58.8 | 28 | 58.3 | 29 | 60.4 |
| Missing | 3 | 2.7 | 1 | 5.9 | 1 | 2.1 | 1 | 2.1 |
| **Alcohol intake** | | | | | | | | |
| Never | 7 | 6.2 | 1 | 5.9 | 1 | 2.1 | 5 | 10.4 |
| Sometimes | 46 | 40.7 | 7 | 41.2 | 21 | 43.8 | 18 | 37.5 |
| Every week | 57 | 50.4 | 8 | 47.1 | 24 | 50.0 | 25 | 52.1 |
| Missing | 3 | 2.7 | 1 | 5.9 | 2 | 4.2 | 0 | 0.0 |
| **Ethnic group** | | | | | | | | |
| White | 107 | 94.7 | 15 | 88.2 | 45 | 93.8 | 47 | 97.9 |
| Black/Asian | 3 | 2.7 | 0 | 0.0 | 2 | 4.2 | 1 | 2.1 |
| Mixed/Other | 1 | 0.9 | 1 | 5.9 | 0 | 0.0 | 0 | 0.0 |
| Missing | 2 | 1.8 | 1 | 5.9 | 1 | 2.1 | 0 | 0.0 |
| **Education** | | | | | | | | |
| No formal qualifications | 16 | 14.2 | 1 | 5.9 | 9 | 18.8 | 6 | 12.5 |
| Secondary education | 49 | 43.4 | 6 | 35.3 | 21 | 43.8 | 22 | 45.8 |
| Higher education | 46 | 40.7 | 9 | 52.9 | 17 | 35.4 | 20 | 41.7 |
| Missing | 2 | 1.8 | 1 | 5.9 | 1 | 2.1 | 0 | 0.0 |
| Household size, median (IQR) | 2.4 | 1.2 | 2.4 | 1.2 | 2.3 | 1.2 | 2.5 | 1.3 |
| Adults, mean (SD) | 2.1 | 0.9 | 2.1 | 0.8 | 2.0 | 0.7 | 2.2 | 1.0 |
| Children, mean (SD) | 0.3 | 0.8 | 0.2 | 0.6 | 0.3 | 0.8 | 0.4 | 0.8 |
| **Weekly grocery shopping, e.g., spending >£25/trip** | | | | | | | | |
| >Once a week | 32 | 28.3 | 3 | 17.7 | 9 | 18.8 | 20 | 41.7 |
| Once a week | 64 | 56.6 | 9 | 52.9 | 31 | 64.6 | 24 | 50.0 |
| Once a fortnight | 6 | 5.3 | 2 | 11.8 | 2 | 4.2 | 2 | 4.2 |
| Once a month | 6 | 5.3 | 1 | 5.9 | 4 | 8.3 | 1 | 2.1 |
| <Once a month | 3 | 2.7 | 1 | 5.9 | 1 | 2.1 | 1 | 2.1 |
| Missing | 2 | 1.8 | 1 | 5.9 | 1 | 2.1 | 0 | 0.0 |
| **Relevant health history** | | | | | | | | |
| Family history | 16 | 14.2 | 4 | 23.5 | 6 | 12.5 | 6 | 12.5 |
| CVD | 4 | 3.5 | 2 | 11.8 | 1 | 2.1 | 1 | 2.1 |
| High blood pressure | 27 | 23.9 | 4 | 23.5 | 14 | 29.2 | 9 | 18.8 |
| Diabetes | 0 | 0.0 | 0 | 0.0 | 0 | 0.0 | 0 | 0.0 |

*(Continued)*

**Table 1.**  (Continued)

|  | Total (n = 113) | | Control (n = 17) | | Brief Support (n = 48) | | Brief Support plus Shopping Feedback (n = 48) | |
|---|---|---|---|---|---|---|---|---|
|  | Mean/n | SD/% | Mean/n | SD/% | Mean/n | SD/% | Mean/n | SD/% |
| **AF** | 4 | 3.5 | 2 | 11.8 | 1 | 2.1 | 1 | 2.1 |
| **CKD** | 1 | 0.9 | 0 | 0.0 | 1 | 2.1 | 0 | 0.0 |
| **Relevant medications** |  |  |  |  |  |  |  |  |
| **Statins** | **2** | **1.8** | **0** | **0.0** | **1** | **2.1** | **1** | **2.1** |

AF, atrial fibrillation; BMI, body mass index; CKD, chronic kidney disease; CVD, cardiovascular disease; IQR, interquartile range.

between groups in total fat, energy density of the basket, sugars, dietary fiber, salt, or basket cost (**Tables C-D in** S4 Appendix).

Purchases of key food groups (%TE) were not statistically significant between the intervention arms versus control, but there were nonsignificant reductions from baseline to follow-up among the intervention groups, which usually paralleled those observed from the dietary questionnaire data: cakes, biscuits, and desserts (BS = −2.7% [95% CI −6.1 to 0.7]; SF = −3.0% [95% CI −6.4 to 0.4]); meat (BS = −2.1% [95% CI −4.4 to 0.2]; SF = −0.2% [95% CI −2.4 to 2.1]); higher fat cheese (BS = −0.4% [95% CI −1.3 to 0.5]; SF = −0.5% [95% CI −1.4 to 0.5]); and higher fat spread (BS = −0.2% [95% CI −1.4 to 1.1]; SF = −0.7% [95% CI −2.0 to 0.5]) (Fig 3, **Tables C-D in** S4 Appendix).

## Changes in clinical outcomes

Changes in blood lipids were not statistically significantly different between intervention and control. In the BS group, there was a nonsignificant reduction from baseline to follow-up in LDL (−0.39 mmol/L [95% CI −0.59 to −0.19]), and in the SF group (−0.14 mmol/L [95% CI −0.34 to 0.07]) (Table 2, **Table E in** S4 Appendix). Changes in body weight and blood pressure were not significantly different between groups either, but there were small nonsignificant

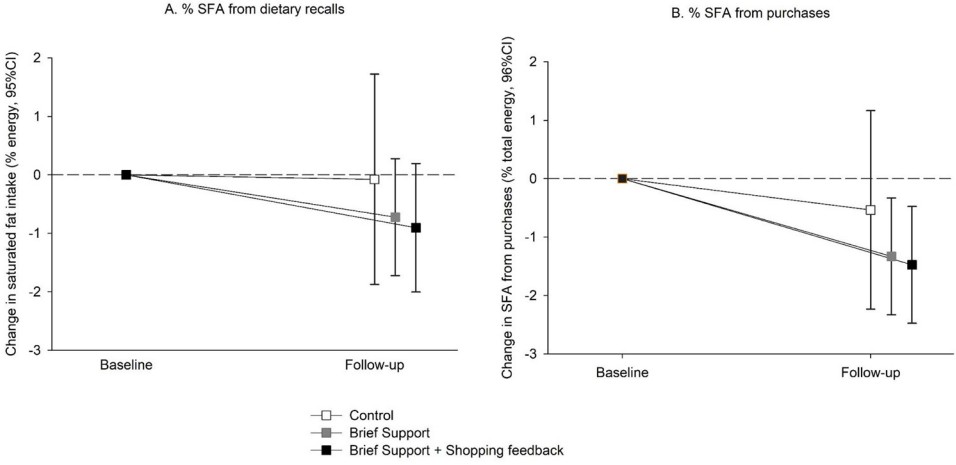

**Fig 2. Mean (± standard error) changes in saturated fat (% energy intake) from A. dietary recalls (primary outcome) and B. purchases (secondary outcome).** SFA, saturated fatty acids.

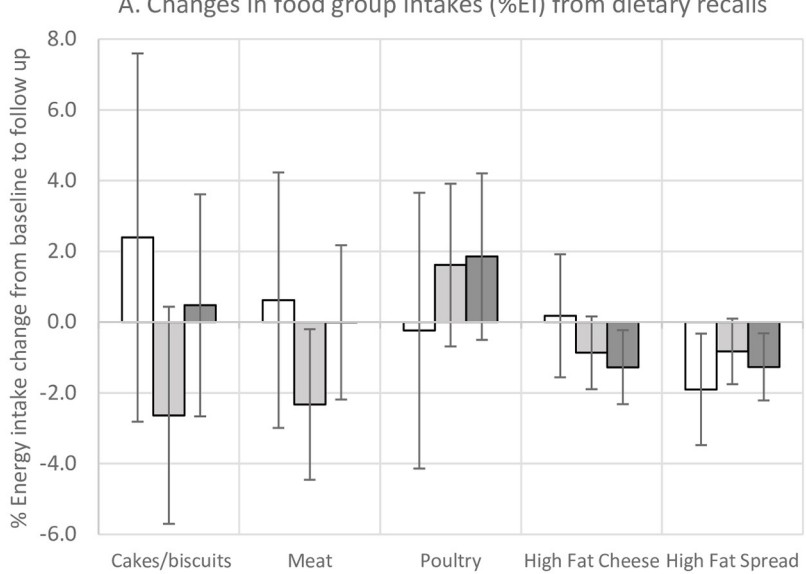

A. Changes in food group intakes (%EI) from dietary recalls

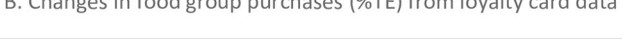

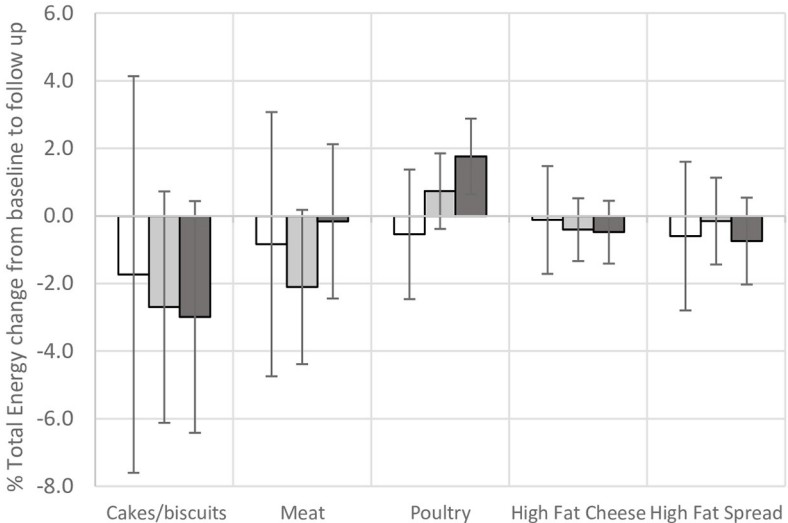

B. Changes in food group purchases (%TE) from loyalty card data

□ Control    ◪ Brief Support    ◼ Brief Support + Shopping Feedback

**Fig 3. Mean (95% CI) changes in key food groups from A. dietary recalls (%EI) and B. loyalty card data (%TE).** EI, energy intake; TE, total energy.

reductions in weight from baseline to follow-up in both intervention groups (BS = −1.1% [95% CI −1.9 to −0.3]; SF = −0.8% [95% CI −1.6 to 0.0]).

## Exploratory subgroup analyses

There was no evidence of a significant interaction between trial arm and education status (secondary education only versus more advanced education) on %SFA intake, %SFA purchases, or

**Table 2. Changes in clinical outcomes: Secondary and exploratory outcomes by group allocation.**

| | Change from baseline within group | | | | | | Between-group differences adjusted for baseline and practice* | | | | | | | | |
|---|---|---|---|---|---|---|---|---|---|---|---|---|---|---|---|
| | Control (*n* = 16) | | BS (*n* = 45) | | BS plus SF (*n* = 44) | | BS versus control | | | BS+SF versus control | | | BS+SF versus BS | | |
| **Secondary Outcomes** | Mean | 95% CI | Mean | 95% CI | Mean | 95% CI | Mean | 95% CI | *p*-value | Mean | 95% CI | *p*-value | Mean | 95% CI | *p*-value |
| LDL cholesterol (mmol/L) | −0.14 | (−0.48, 0.19) | −0.39 | (−0.59, −0.19) | −0.14 | (−0.34, 0.07) | −0.15 | (−0.47, 0.16) | 0.338 | 0.04 | (−0.28, 0.36) | 0.790 | 0.2 | (−0.03, 0.43) | 0.095 |
| HDL cholesterol (mmol/L) | 0.08 | (−0.03, 0.19) | −0.01 | (−0.08, 0.05) | −0.02 | (−0.09, 0.04) | −0.11 | (−0.24, 0.01) | 0.081 | −0.13 | (−0.26, 0.00) | 0.047 | −0.02 | (−0.11, 0.07) | 0.713 |
| Total cholesterol (mmol/L) | −0.13 | (−0.46, 0.21) | −0.46 | (−0.66, −0.25) | −0.14 | (−0.35, 0.06) | −0.3 | (−0.62, 0.02) | 0.069 | −0.02 | (−0.34, 0.30) | 0.910 | 0.28 | (0.04, 0.51) | 0.02 |
| Triglycerides (mmol/L) | −0.14 | (−0.54, 0.26) | −0.14 | (−0.38, 0.11) | −0.01 | (−0.25, 0.24) | 0.04 | (−0.37, 0.46) | 0.836 | 0.27 | (−0.15, 0.69) | 0.211 | 0.22 | (−0.09, 0.53) | 0.154 |
| Non−HDL cholesterol (mmol/L) | −0.21 | (−0.56, 0.14) | −0.44 | (−0.65, −0.24) | −0.12 | (−0.33, 0.09) | −0.16 | (−0.51, 0.19) | 0.362 | 0.15 | (−0.20, 0.50) | 0.393 | 0.31 | (0.06, 0.56) | 0.016 |
| Total cholesterol/HDL ratio | −0.15 | (−0.59, 0.29) | −0.34 | (−0.60, −0.07) | 0.02 | (−0.24, 0.29) | −0.13 | (−0.64, 0.37) | 0.601 | 0.25 | (−0.26, 0.76) | 0.335 | 0.38 | (0.01, 0.75) | 0.042 |
| **Nonefficacy Outcomes** | | | | | | | | | | | | | | | |
| Systolic blood pressure (mmHg) | 0.5 | (−6.3, 7.4) | 1.3 | (−2.8, 5.4) | 1.2 | (−3.0, 5.3) | 0.58 | (−7.01, 8.18) | 0.879 | 1.44 | (−6.23, 9.11) | 0.710 | 0.86 | (−4.68, 6.40) | 0.759 |
| Diastolic blood pressure (mmHg) | 1.2 | (−2.7, 5.2) | −0.3 | (−2.7, 2.0) | −0.5 | (−2.9, 1.9) | −1.42 | (−5.80, 2.97) | 0.523 | −0.01 | (−4.50, 4.48) | 0.997 | 1.41 | (−1.87, 4.69) | 0.397 |
| Weight (kg) | −0.2 | (−1.5, 1.1) | −1.1 | (−1.9, −0.3) | −0.8 | (−1.6, 0.0) | −1.00 | (−2.52, 0.53) | 0.197 | −0.57 | (−2.11, 0.98) | 0.468 | 0.43 | (−0.69, 1.55) | 0.449 |

* Estimates from linear regression models adjusting for GP practice and baseline values, all *p*-values > 0.05.

BS, brief support; GP, general practioner; HDL, high-density lipoprotein; LDL, low-density lipoprotein; SF, brief support plus shopping feedback.

LDL-C (Fig 4, **Table F in** S4 Appendix). Among those allocated to BS group but not SF group, there was a greater reduction in SFA consumption and purchase in participants with lower levels of education, but that was not statistically significant to control (Fig 4).

## Sensitivity analyses

None of the sensitivity analyses changed the overall pattern of results but in most cases, resulted in slightly larger absolute differences between the intervention groups and control group (**Table G in** S4 Appendix).

## Process evaluation: Feasibility, intervention fidelity, and acceptability

Of 4,766 people invited to take part in the study, 181 (4%) responded to the invitation, from which 143 participants (3%) fulfilled some of the eligibility criteria (e.g., were customers of the participating supermarket) and consented and agreed to be screened for full eligibility. Of those, 30 (21%) were ineligible, mostly because LDL cholesterol was <3 mmol/L at the time of recruitment, giving 2.4% of the original sample of invited participants who were then randomized. Of these, 94% were followed up after 3 months.

There was an average of 37 days (SD 22) between the baseline visit with researchers and the brief advice appointment with the HCP. The BS sessions delivered by the HCP were audio-recorded with an average duration of 10 minutes (range 3 minutes to 30 minutes). Out of 19 possible behavioral elements included in the training, an average of 10 (range 3–19) were evident in the recordings. Two of the elements that were missed the most were asking participants

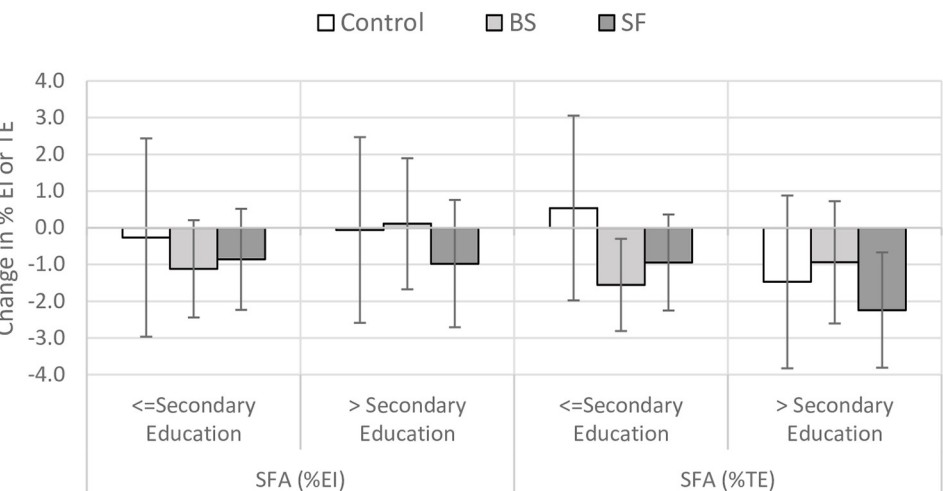

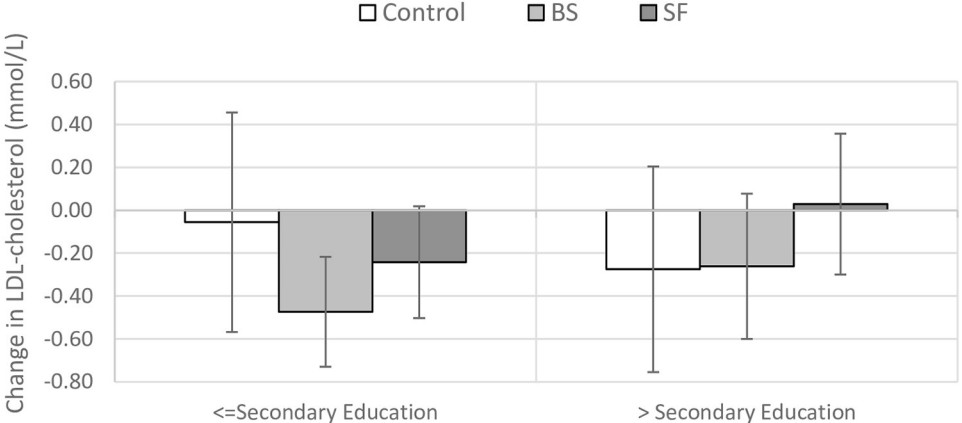

**Fig 4. Mean (95% CI) changes in saturated fat from A. dietary recalls (%EI) and loyalty card data (%TE) and B. LDL cholesterol (mmol/L) by group and education status.** BS, brief support; EI, energy intake; LDL, low-density lipoprotein; SF, brief support plus shopping feedback; SFA, saturated fatty acids; TE, total energy.

if they had already identified a specific food to change and the take-home message in which the HCP provided an overall overview of the main points discussed.

The SF group received a mean of 3.6 shopping reports. Each shopping report included 5 potential swaps. Approximately 41% of lower SFA swaps offered were dairy products (e.g., cheese, yogurt, milk/cream, spread/butter); 21% cakes, biscuits, and desserts; 17% meat (e.g. beef, pork, processed meat); 7% ready meals (e.g., pizza, pies); and 13% from other categories (e.g., salty snacks, oils, bread, cereal). One in 4 suggested swaps appeared in the subsequent purchase record during the intervention period. These "accepted swaps" came from cheese, yogurt, milk/cream or spread/butter (58%); cakes, biscuits, and desserts (15%); meat (13%); salty snacks (2%); and ready meals (2%) (**Table H in** S4 Appendix).

Overall, participants reported that the HCP intervention session and self-help materials were helpful, with mean scores of 3.9 among BS and 3.7 among SF (on a scale 1–5).

Participants in the SF group considered the shopping report was moderately helpful with a score of 3.5, and 43% reported using the shopping report "almost always" or "most of the time" when shopping. A higher proportion of participants in the intervention groups reported being "very/somewhat confident" that they knew the major sources of SFA in their diets after the intervention. A higher proportion of participants in all groups were trying to reduce SFA in their diets at follow-up than at baseline (**Table H in** S4 Appendix).

## Discussion

Among people identified as having modestly raised LDL-C, there was no strong evidence that a brief behaviorally informed appointment with a HCP with or without additional personalized shopping feedback and swap suggestions for the saturated fat content of food purchases reduced SFA consumption, SFA food purchases, or LDL-C relative to a control group. However, the reductions in SFA and LDL cholesterol in the intervention groups were not significant but larger than those in the control with no formal intervention. The intervention was feasible, acceptable, and potentially scalable.

This pragmatic study did not detect a reduction in SFA or LDL cholesterol of a magnitude to imply that this intervention is an effective alternative to lipid-lowering therapy for individuals with raised LDL. Explanatory trials, which provided specific foods to replace higher SFA products, together with intensive tailored counseling, have shown greater reductions in SFA (approximately 4%–5%) with small reductions in LDL cholesterol of approximately 0.2 mmol/L [8] and other CVD risk factors over 12–14 weeks of intervention [8, 9]. However, our data are consistent with a previous pilot trial of dietary counseling for people with dyslipidaemia in primary care that showed a similar small reduction in LDL (−0.32 mmol/L, $p < 0.05$ in the active intervention arm at 6 months compared with baseline) [31].

Although there is limited clinical benefit to individuals of very small reductions in SFA, there could be important population-level health gains. Systematic reviews of trials of interventions to reduce SFA intake compared with usual diets showed interventions reduced LDL by −0.19 mmol/L (−0.33, −0.05) yet decreased the incidence of nonfatal CVD events by 17% [3], and modeling suggests that just replacing 1% of EI from SFA with PUFA reduces CVD events by 8% [32]. In testing this novel intervention, we opted for a small "hybrid" trial examining the feasibility of the shopping feedback and looking for evidence of effectiveness as a clinical intervention, recognizing that it would be underpowered to detect important population effects [22]. The process measures suggest further research on the shopping feedback delivered at a larger scale is now justified.

A strength of this study is the randomized controlled trial (RCT) design and high follow-up rate (94%), which reduces the effect of confounding and provides high internal validity. The average monthly spend across groups and over time was largely unchanged, suggesting that the intervention did not disrupt usual shopping patterns. Our primary outcome was based on a dietary recall method, which is subject to misreporting, and included only 2 days of reporting at baseline and follow-up, but potential bias is reduced by the comparison with a control group. Total EI data measures are consistent with the observed weight changes, and there was good agreement in the measured change in SFA and key food groups in each group between the dietary recall and the household food purchase data. A previous study also showed strong correlations approximately 0.8 between total energy and fat between dietary recalls and food purchases, which suggest that the use of purchasing data is feasible, inexpensive, and accurate for nutritional research in populations in which most food comes from supermarkets [29].

The principal limitation of the study is the sample size, which lacked power to detect smaller effect sizes. The participants randomized were a very small proportion of those eligible

and may have been biased to those with higher levels of concern about their diet. It is possible that participants in the control group could have been prompted to reduce their SFA upon receiving their blood test results and the notice of a future check 3 months later, which reduced the chance of detecting a difference between groups. Only 2 participants were taking lipid-lowering medications, and this may also imply a motivated group already taking action, which could limit the opportunity for further change. Our population included a higher proportion of middle/older age participants, reflecting the population with dyslipidaemia [33]. Women were over-represented, but this probably reflects the requirement to be the main shopper in the household to participate in the study. The recruitment of participants was also limited to people whose weekly shopping was mostly from the participating supermarket. However, the collaborating supermarket we worked with accounts for 27% of the UK grocery market, with customers across the socioeconomic spectrum. A future trial recruiting a larger sample of participants would require a larger sample of GP practices in order to recruit at a satisfactory rate if similar inclusion criteria are used. The limited fidelity in the HCP-led intervention needs to be acknowledged, because only half of the intended behavioral intervention components were delivered on average. In an explanatory trial, this would be problematic, but in this pragmatic trial, lower fidelity is not inherently a limitation. However, it suggests that further attention needs to be given to the training for HCPs in any future trial. Likewise, it would be ideal to develop a system to ensure that participants randomized to the intervention arms were able to see the HCP sooner than occurred here to capitalize on initial enthusiasm. A sensitivity analysis, which excluded people who received their intervention after 45 days from recruitment, revealed slightly stronger though still nonsignificant differences between intervention groups and control.

The shopping feedback used a novel computerized algorithm to process information from loyalty cards and provide tailored dietary advice in real-time. By drawing on the power of data already collected by grocery stores, we could deliver personalized advice, which would otherwise require specialist practitioners, usually dieticians, considerable time to analyze and would strain healthcare resources. There was no difference observed between the groups receiving brief advice only and those also receiving personalized shopping reports. For this highly selected population, it is possible that brief advice is enough to change food purchasing and consumption, and we do not know the effect of tailored shopping advice alone. Only 3 studies have previously observed absolute reductions in SFA intake with interventions focused on food purchasing. A study of tailored education (e.g., frequently mailed tailored shopping lists, recipes, and healthier alternatives) based on previous purchases showed a reduction of −1.4% in SFA in all food purchases at 6 months, but this was not significantly greater than the control group [34]. The amount of SFA in purchases also decreased by 0.7 g SFA/week in a pilot study providing web-based advice based on the traffic-light labeling system to improve the quality of ready meals (e.g., pizza) [35]. Only one study, performed in an online supermarket, which offered lower SFA swaps on food products (excluding meat), achieved a significant difference between the intervention and control group of −0.66% SFA [19].

Importantly saturated fat intake and %SFA in total purchases declined in both intervention groups for the food categories that contribute the most to SFA intakes in the UK, and the changes were congruent with the emphasis of the advice provided. The written information provided in the brief advice group focused on meat as well as dairy intake and purchases and intake of these foods, especially meat, declined in the BS group. In contrast, most swaps offered in the shopping reports and accepted in the SF group were for dairy products. The slightly larger reductions in LDL cholesterol in the BS compared with the SF group are consistent with recent epidemiological evidence on the effects of individual fatty acids found in different concentrations in meat and dairy products [36].

In qualitative questionnaires, participants highly valued the shopping reports and reported them to be useful for self-monitoring. Given the relative ease and low cost of providing this information at scale, it warrants further investigation as a public health intervention for people seeking to reduce the SFA content of the diet. A similar approach could potentially be adopted for other nutrients of concern, e.g., salt or free sugars, by retail partners who may wish to offer this type of support to their customers. Only a few studies have used information on household food purchases from loyalty cards, usually to monitor the effectiveness of interventions or to provide personalized dietary advice [34, 35, 37–39], but none have developed a scalable system as we used here to provide this information in real-time. At present, access to this data requires complex legal agreements with the retailers to address the privacy concerns of consumers and commercial concerns of retailers [40]. However, in an era of open data, greater awareness by the "data creators" (i.e., their customers) that the "data holders" (i.e., companies) are required to share this information on request might persuade companies to set up routine systems to use these data in ways that add value to the consumer experience and also promote healthier purchasing.

In conclusion, our trial showed nonsignificant reductions in SFA intake, the SFA content of household food purchases, and LDL-C. The small changes we observed are consistent with other individual-level behavioral interventions to reduce SFA, but the study was underpowered to detect all but large improvements. With further enhancement of the intervention based on participants' feedback, the good evidence of feasibility and acceptability of using grocery loyalty card data offers the potential to develop an automated, low-cost, and scalable intervention that could reach and benefit large numbers of people to deliver population-level impacts, which may improve public health.

## Supporting information

**S1 Appendix. Study protocol.**
(DOCX)

**S2 Appendix. List of inclusion and exclusion criteria.**
(DOCX)

**S3 Appendix. Qualitative questionnaire.**
(DOCX)

**S4 Appendix. Supplementary tables.**
(DOCX)

**S5 Appendix. CONSORT checklist.** CONSORT, Consolidated Standards of Reporting Trials.
(DOC)

## Acknowledgments

We would like to thank Mei-Man Lee (medical statistician) for providing input into the statistical plan and the sample size calculation for this trial. We would like to thank the members of the public who helped reviewing materials at the initial stages of the grant application and during the intervention development. We also thank the 4 practices that participated in the study and the practice staff (practice managers, HCP, GPs, receptionists) involved in the recruitment and intervention delivery. We finally thank all the participants that were willing to take part in the study.

The views expressed in this publication are those of the authors and not necessarily those of the National Institute for Health Research or the Department of Health and Social Care.

## Author Contributions

**Conceptualization:** Carmen Piernas, Paul Aveyard, Nerys M. Astbury, Susan A. Jebb.

**Data curation:** Carmen Piernas, Charlotte Lee, Jason Oke.

**Formal analysis:** Jason Oke.

**Funding acquisition:** Carmen Piernas, Paul Aveyard, Susan A. Jebb.

**Investigation:** Carmen Piernas, Charlotte Lee, Melina Tsiountsioura, Michaela Noreik, Claire Madigan, Susan A. Jebb.

**Methodology:** Carmen Piernas, Paul Aveyard, Charlotte Lee, Melina Tsiountsioura, Michaela Noreik, Nerys M. Astbury, Jason Oke, Claire Madigan, Susan A. Jebb.

**Project administration:** Carmen Piernas, Charlotte Lee, Melina Tsiountsioura, Michaela Noreik, Nerys M. Astbury, Claire Madigan.

**Supervision:** Carmen Piernas, Paul Aveyard, Susan A. Jebb.

**Writing – original draft:** Carmen Piernas.

**Writing – review & editing:** Carmen Piernas, Paul Aveyard, Charlotte Lee, Melina Tsiountsioura, Michaela Noreik, Nerys M. Astbury, Jason Oke, Claire Madigan, Susan A. Jebb.

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
