## [Decision Letter · Decision Letter 0]

31 Jan 2020

Dear Dr. Piernas,

Thank you very much for submitting your manuscript "Brief support and personalised feedback on food shopping to encourage saturated fat reduction: results from the PC-SHOP randomised controlled trial" (PMEDICINE-D-19-03921) for consideration at PLOS Medicine. 

Your paper was evaluated by a senior editor, discussed with an academic editor with relevant expertise, and sent to independent reviewers, including a statistical reviewer. The reviews are appended at the bottom of this email and any accompanying reviewer attachments can be seen via the link below:

[LINK]

In light of these reviews, I am afraid that we will not be able to accept the manuscript for publication in the journal in its current form, but we would like to consider a revised version that addresses the reviewers' and editors' comments. Obviously we cannot make any decision about publication until we have seen the revised manuscript and your response, and we plan to seek re-review by one or more of the reviewers. 

We expect to receive your revised manuscript by Feb 14 2020 11:59PM. Please email us (plosmedicine@plos.org) if you have any questions or concerns.

We look forward to receiving your revised manuscript. 

Sincerely,

Louise Gaynor-Brook, MBBS PhD

Associate Editor 

PLOS Medicine

plosmedicine.org

General comment: Please add a space before reference brackets 

Data Availability Statement: "PLOS Medicine requires that the de-identified data underlying the specific results in a published article be made available, without restrictions on access, in a public repository or as Supporting Information at the time of article publication, provided it is legal and ethical to do so. Please see the policy at http://journals.plos.org/plosmedicine/s/data-availability and FAQs at 

http://journals.plos.org/plosmedicine/s/data-availability#loc-faqs-for-data-policy

If the data are not freely available, as you have stated, please describe briefly the ethical, legal, or contractual restriction that prevents you from sharing it. Please also include an appropriate contact (web or email address) for inquiries (please note that this cannot be a study author)

Please revise your title according to PLOS Medicine's style, placing the study design in the subtitle (ie, after a colon). We suggest “Evaluation of an intervention to provide brief support and personalised feedback on food shopping and saturated fat intake (PC-SHOP): a randomised controlled trial” or similar. 

Please report your abstract according to CONSORT for abstracts, following the PLOS Medicine abstract structure (Background, Methods and Findings, Conclusions) http://www.consort-statement.org/extensions?ContentWidgetId=562

Abstract Background: Please expand upon the context of why the study is important. 

Abstract Methods and Findings:

Please include brief demographic details of the participants (e.g. age, sex, etc) and further details of the study setting (e.g. locations), number of participants, dates between which the study took place and length of follow up.

Please state that analysis was intention to treat.

Please quantify the main results (with 95% CIs and p values).

In the last sentence of the Abstract Methods and Findings section, please describe the main limitation(s) of the study's methodology.

Please begin your Abstract Conclusions with “"In this study, we observed ..." or similar. Please address the study implications, emphasizing what is new without overstating your conclusions.

At this stage, we ask that you include a short, non-technical Author Summary of your research to make findings accessible to a wide audience that includes both scientists and non-scientists. The Author Summary should immediately follow the Abstract in your revised manuscript. This text is subject to editorial change and should use non-identical language distinct from the scientific abstract. Please see our author guidelines for more information: https://journals.plos.org/plosmedicine/s/revising-your-manuscript#loc-author-summary

Introduction

Please further address past research and explain the need for and potential importance of your study. If there has been a systematic review of the evidence related to your study (or you have conducted one), please refer to and reference that review and indicate whether it supports the need for your study. 

Line 80 - please add ‘to our knowledge...’ or similar, to avoid assertions of primacy.

Methods 

Please complete the CONSORT checklist and ensure that all components of CONSORT are present in the manuscript. When completing the checklist, please use section and paragraph numbers, rather than page numbers

With regard to your prospective protocol and analysis plan, please make sure that the Methods section transparently describes when analyses were planned, and whether or not any data-driven changes to analyses took place (indicating what those are)

Line 96 - please clarify whether informed consent was written or oral 

Line 102 - please specify how many primary care practices were involved 

Line 209-212 - please expand on the methods used to assess these outcomes

Line 214 - please define SD (to avoid confusion with standard deviation)

Line 221 - please clarify what is meant by ‘ 2% (3% SD)’

Line 230 - Please provide a copy of the questionnaires used in your supplementary information. 

Results

Please provide p values in addition to 95% CIs throughout your results section.

Please include Figure S1 in the main text of your manuscript. 

Please define the length of follow up (eg, in mean, SD, and range).

Line 261 - Please indicate which factors are adjusted for

Line 275 - please revise ‘somewhat greater reductions’ to clarify that reductions were non-significant 

Line 306 - please revise ‘slightly larger reductions’ to clarify that reductions were non-significant 

Line 356 - please replace % with ‘percentage’, ‘proportion’ or similar 

Discussion

Line 369 - please revise ‘somewhat greater reductions’ to clarify that reductions were non-significant 

Line 450 - please clarify which groups are being compared when discussing these results 

Line 482 - please avoid using the word ‘novel’, in order to avoid assertions of primacy 

Figure 1 - please define SFA and SEM in the figure legend 

Figure 2 - please define EI and TE in the figure legend 

Figure 3 - please define EI, TE, SFA, BS and SF, LDL in the figure legend 

Supplementary Tables - please define all abbreviations used in the table legend 

Table S6 - When a p value is given, please specify the statistical test used to determine it.

References

Please ensure that all references are appropriately formatted and capitalised e.g. refs 7, 25, 27, 28, 29, 31, 33

Comments from the reviewers:

Reviewer #1: Thank you for the opportunity to review your manuscript. This study presents the findings of a 3-arm RCT that tests whether there are changes in saturated fat intake (% total energy) when patients with high LDL cholesterol are exposed to usual care, brief support, and brief support with enhanced feedback on shopping purchases. My role is as a statistical reviewer and I have focused on the design and analysis. The protocol and statistical analysis plan are included, the supplementary material attached to this publication is well-organised and I found it helpful when conducting my review. I have previously collaborated with colleagues working in general practice, primary prevention, and food databases and I appreciate the thoughtfulness and potential of this work. 

P6, Line 114. A 1:3:3 allocation ratio is used, but there doesn't appear to be a justification for this in the protocol or manuscript. Was there a specific reason that the usual practice control group had a smaller allocation? 

P6, Line 116. The protocol (p21) states that randomly varying block sizes were used, but then specifies that all the blocks were of size 7. Can you confirm whether they were randomly sized, or used a fixed block size with random order?

P10, L213. 'Statistically significant' is used throughout the reporting of results, but there are not details as to what was statistically significant in the methods (e.g. alpha < 0.05, 95% CI overlaps null effect?). If 'statistically significant' is to be used, then this should be formally defined, if based on a p-value then this should be presented. 

P10, L214 Where was the standard deviation for % of SFA for the sample size calculations taken from?

P10, L219. In the power calculation it says that a multiciplicy adjustment is included, but there are no details of what the selected multiplicity adjustment was that was incorporated into the sample size calculation, and whether a multiplicity adjustment was applied to the final analysis (and what it was). 

P13, Table 1. Just confirming the results for BMI by category vs. average BMI. There is very little difference between groups in average BMI, but when reported as % per categories there appears to be some variation between groups. 

P21, L226. The primary analysis not only includes an adjustment for baseline value, but also for practice. Just to clarify, was the practice included as a fixed effect or as a random effect in the main regression model? 

P22, L318. The most powerful test of differences between subgroups showed no important differences between groups, but the last sentence gives the impression that there was a useful difference found - I think the wording should be adjusted to reflect this. Also, if a p-value is used as evidence for or against an interaction, it should be presented.

P24, L326. The results of the sensitivity analyses should be included in the supplementary materials.

Consort checklist: Have the page numbers reported in the checklist shifted when it was formatted for submission? I couldn't find some of the items (e.g. eligibility and trial design should be on p4 and p3) in the manuscript in the pages they are reported.

Reviewer #2: 

Thank you for the opportunity to read this interesting paper reporting a well-conducted 3-arm hybrid trial examining changes in SFA intake after a behavioural intervention. The retention rate is excellent and the finding that no significant between-group differences in changes emerged over this timescale in this particular sample size is an important one to inform future work. I have some suggestions to further clarify aspects of the manuscript, which the authors may like to consider.

1. Typo in short title - 'amd' should be 'and'

Abstract

2. Would the authors consider mentioning that that RCT was a hybrid trial in the Abstract? Also, in the abstract it may help the reader to mention the retention rate / note the final N and to note that the primary outcome came from self-reported 24hr recalls of intake.

3. The values for adjusted between group differences do not appear to be reported in the Abstract, although this is stated as the primary outcome in the Abstract itself. Unless I have misunderstood this, these adjusted values should be reported as well as - or instead of - the absolute, unadjusted % SFA intake changes currently reported.

Method

4. Were outcome assessors of the variables that could not be reported via online portal blinded to allocation - for example, body weight? I assume that this is what is referred to when talking about blinding at follow up, but this could benefit from clarification.

5. I found myself searching for some of the information contained in supplementary material in order to understand the basics of the RCT. A key example of this was participant exclusion/inclusion based on sufficient use (or not) of the specific supermarket chain. Is there scope to include more details about inclusion/exclusion in the main body of the Method?

6. Could data for the numbers of participants who did not respond to the invitation / who were screened ineligible prior to consent be added to the CONSORT diagram at the first stage - this is not essential but provides additional interesting information that is presently presented in the text?

7. Please consider whether it would be appropriate to provide brief extra behavioural details of the intervention delivered (e.g. 'box 1' as shown in Lombard et al, 2016; https://journals.plos.org/plosmedicine/article?id=10.1371/journal.pmed.1001941). I realise that details of the intervention are provided in the protocol paper, but a very brief summary would be immensely helpful to behaviourally-oriented readers.

8. Please could the rationale for presenting both 24 hr recall and supermarket data be more clearly stated?

Results

9. Please consider whether figure and table notes need to define acronyms like SFA and EI. 

10. As it took on average 37 days from baseline to intervention delivery, would the authors also consider reporting the mean number of days between receiving the intervention and completing follow-up assessment? Did the 3 month period start from the day of the baseline assessment or the day of the intervention appointment? How did this month-long average delay affect the number of shopping feedback components delivered to participants in this group - did all participants receive all 4 sets of feedback (as is suggested by an average of 3.6 in the SF group)?

11. Line 274: "Intakes of key food groups (%EI) were not statistically significant between intervention and control" - should 'significant' read 'different'; the sentence does not quite scan?

12. Were any particular behavioural components omitted from delivery more frequently than others? It would be interesting to know a little more about the outcome of the fidelity analyses but not essential.

Discussion

13. "However, the reductions in SFA and LDL-cholesterol in the intervention groups were somewhat greater than those in the control with no formal intervention" - can this claim be modified to remind the reader that such changes (and, later in the Discussion, the change in SFA in purchases) did not reach statistical significance?

14. Might the relatively brief duration of the intervention period be a study limitation?

15. Given that the trial also assessed the feasibility of the recruitment process, what changes would the team consider to this process given the relatively low response rate of 3% of potentially eligible participants?

Abbreviations

16. Might EI and TE be profitably added to the list of abbreviations?

References

17. Is there an issue with reference 8 (Office of national statistics?) Similarly references 20 and 21. References 33 and 34 seem to be duplicates.

Reviewer #3: Thank you for the opportunity to review this paper. 

This is a small randomized trial (n=113) to 3 groups a Brief Support a Brief support +Shopping feedback and a control group that reports feasibility but no effectiveness. This is an interesting intervention, potentially scalable and a novel idea which needs evaluation, however this is underpowered and can only really establish the feasibility of implementing this strategy and not the effectiveness or otherwise of this strategy. Important partnerships have been established to enable this and likely work done to develop the intervention. Overall this is well written, but I think there could be on balance more emphasis on the feasibility/ implementation aspects and less on the effectiveness. 

Methods: 

Why did the investigators only target patients with high LDL in the last 5 years? An Absolute risk approach could have been taken, or it could have been justified to target a larger population group as a saturated fat intervention is likely to be relevant to a wider group of patients. Why were patients with CVD excluded? A dietary intervention on-top of there usual treatment would seem reasonable. And the primary outcome was self-reported saturated fat and not LDL cholesterol. 

Why was a 1:3:3 ratio used? 

Research staff not being blinded to allocation could have been an issue, but the primary outcome was assessed through an electronic questionnaire without involving the study team. 

It would be good to describe here or in discussion about the development of the partnership with the UK grocery store in more detail and the design of the shopping report, who was involved in the design and was it tested with consumers prior to the trial. I think this might add more depth to this work.

Results: 

There seems to be some imbalances in some risk factors (e.g. normal weight), this is likely to have been chance given the small sample size of groups, however it would be helpful to see what the distribution of absolute cardiovascular risk was across participants. 

Discussion:

This is generally well discussed. There is some repetition and could probably be trimmed if word count/ length an issue. The focus of conclusions is effectiveness but probably the focus should be feasibility and implementation evaluation. It would be good to emphasize and discuss the learnings from the intervention implementation. 

The issue of not all the behavioural components being delivered is probably a challenge of implementation and it may be good to discuss how this may be addressed in the future, is it just the training? Or is the HCP step needed? Or are there other ways of delivering this? Or does the content of delivery matter so much as the contact with the HCP?

In addition: 

I would usually think good to include the inclusion/ exclusion criteria and consort diagram in main paper as opposed to appendix. 

I think a figure to show an example of the shopping report would also be helpful to visualize what this comprised. 

I think having a consort diagram and figure to show the shopping report would be more helpful than existing figures. E.g. perhaps Figure 1 could be put in appendix (this repeats data in text/tables doesn't it), figure 2 could be kept and figure 3 could be put in appendix.

[LINK]

---

## [Decision Letter · Decision Letter 1]

19 Aug 2020

Dear Dr. Piernas,

Thank you very much for re-submitting your manuscript "Evaluation of an intervention to provide brief support and personalised feedback on food shopping to reduce saturated fat intake (PC-SHOP): a randomised controlled trial" (PMEDICINE-D-19-03921R1) for review by PLOS Medicine.

I have discussed the paper with my colleagues and the academic editor and it was also seen again by reviewers. I am pleased to say that provided the remaining editorial and production issues are dealt with we are planning to accept the paper for publication in the journal.

[LINK]

We look forward to receiving the revised manuscript by Aug 26 2020 11:59PM. 

Sincerely,

Adya Misra, PhD

Senior Editor 

PLOS Medicine

plosmedicine.org

Requests from Editors:

Abstract

GP – perhaps not commonly used outside the UK so I would recommend introducing on first view 

Please provide brief participant demographics-include age range, sex etc in the abstract

Please provide p-values as needed. Al p-values should be exact numbers, unless p<0.001

Data statement: We understand the data restrictions due to the ethics and contractual obligations. However, please note we do not permit authors to act as gatekeepers to the primary underlying data. Please provide an institutional or ethics committee contact instead. Regarding the Brand View dataset, please provide details of the dataset and contact details of someone at BrandView who can be approached by interested parties to gain access to the data. 

Author summary

Please add bullet points

Nurse or healthcare professional seems a bit redundant. Could you perhaps just say healthcare professional?

I suggest rephrasing “we followed them for approximately 3 months” as this could mean any number of things

Methods

All references to questionnaires used in the study should be accompanied by a citation to the reference, either as published or as provided in SI files

Line 277- suggest revising “results were averaged…”

CONSORT flowchart-please specify reasons for ineligibility 

Results

Line 375 and line 397 onwards- please provide all p-values to accompany 95% CI

In all instances where you mention “significance” or “non-significance” please provide 95% CI and p values. 

I suggest toning this down “Participants in the control group appeared to have been prompted to reduce their SFA upon receiving their blood test results and the notice of a future check three months later, which reduced the chance of detecting a difference between groups” 

CONSORT checklist- please use paragraphs and sections instead of page numbers as these are likely to change

Overall- please avoid the use of the phrase “somewhat greater” and use significant or non-significant instead. 

Comments from Reviewers:

Reviewer #1: Thanks for the revisions to the manuscript and the details given in the response to my comments (and the other reviewers). The comments and the revisions to the manuscripts have resolved all my original queries. 

One comment that may be covered during further editorial process was that the figures have some issues - in my PDF some of the axis label text is obscured by background stippling and the text isn't rendered properly making it difficult to read. This could be something induced by the MS submission system.

For Figure 3 I'd recommend representing the % change as a dot or square rather than a bar.

Reviewer #2: I'd like to thank the authors very much for their response to my comments and those of the editor and other reviewers - the manuscript is much clearer. The response to comment number 15 in my review regarding engagement from colleagues in primary care in the recruitment process is particularly interesting and illuminating, and very much highlights the need for this sort of study!

I have just one other suggestion for the revised manuscript.

Box describing intervention: Could the description of the shopping report be clarified further please? It currently appears as if the 'report' in point 1 is separate from the information listed in point 2, whereas my understanding is that the report consisted of the information under point 2. Apologies if I have misunderstood. In addition, also in this box, is there any possibility that the number of reports could be specified alongside the statement that they were provided monthly from baseline onwards?

[LINK]

---

## [Editor Report · Decision Letter 2]

18 Sep 2020

Dear Dr. Piernas, 

On behalf of my colleagues and the academic editor, Dr. Barry M. Popkin, I am delighted to inform you that your manuscript entitled "Evaluation of an intervention to provide brief support and personalised feedback on food shopping to reduce saturated fat intake (PC-SHOP): a randomised controlled trial" (PMEDICINE-D-19-03921R2) has been accepted for publication in PLOS Medicine. 

PRODUCTION PROCESS

PRESS

PROFILE INFORMATION

Thank you again for submitting the manuscript to PLOS Medicine. We look forward to publishing it. 

Best wishes, 

Adya Misra, PhD

Senior Editor 

PLOS Medicine

plosmedicine.org